# Optimization of Poly(l-Amino Acids)-Based Platforms for Sensing and Biosensing: A Cyclic Voltammetry Study

**DOI:** 10.3390/s25237230

**Published:** 2025-11-27

**Authors:** Giulia Selvolini, Agnese Bellabarba, Costanza Scopetani, Carlo Viti, Tania Martellini, Alessandra Cincinelli, Giovanna Marrazza

**Affiliations:** 1Department of Chemistry “Ugo Schiff” (DICUS), University of Florence, Via della Lastruccia 3, 50019 Sesto Fiorentino, FI, Italy; costanza.scopetani@unifi.it (C.S.); tania.martellini@unifi.it (T.M.); alessandra.cincinelli@unifi.it (A.C.); giovanna.marrazza@unifi.it (G.M.); 2Laboratory of Phenomics, Genomics, and Proteomics (GENEXPRESS), University of Florence, Via della Lastruccia 14, 50019 Sesto Fiorentino, FI, Italy; agnese.bellabarba@unifi.it (A.B.); carlo.viti@unifi.it (C.V.); 3Department of Agriculture, Food, Environmental and Forestry Sciences (DAGRI), University of Florence, Piazzale delle Cascine 18, 50144 Florence, FI, Italy; 4Center for Colloid and Surface Science (CSGI), University of Florence, Via della Lastruccia 3, 50019 Sesto Fiorentino, FI, Italy

**Keywords:** poly(l-amino acids), gold nanoparticles, screen-printed cells, nanocomposite platform

## Abstract

Poly(amino acids) and gold nanoparticles are stable and biocompatible materials with distinguishing features which can be used to build nanocomposite electrochemical platforms for sensing applications. This paper presents the optimization of the building steps of these nanocomposite platforms using cyclic voltammetry. Screen-printed graphite electrodes were first modified by electropolymerizing various l-amino acids and then by electrodepositing gold nanoparticles. The electroactive surface area was calculated for all platforms, which were then applied in the electrochemical oxidation of 1-naphthol as a model analyte: oxidation peaks were observed in all cases, with the current peak height increasing with increasing analyte concentration, thus demonstrating the potential of nanocomposite platforms for developing electrochemical sensors and biosensors.

## 1. Introduction

Poly(amino acids) (PAAs) are a biocompatible and biodegradable alternative to classical conducting polymers which have attracted much attention in recent years due to a set of advantages, first of all their non-toxicity [1,2,3]. Stable and uniform PAAs films with good adhesion properties can be obtained by simply electropolymerizing the monomers directly onto electrodic surfaces: these films can significantly improve the conductivity of the electrode and enhance the electroactivity of the analytes, while the presence of numerous active sites allows the PAAs to act synergistically with specific materials to improve the analytical performance of electrochemical sensors [4,5,6]. Moreover, these biomaterials can self-assemble into ordered and stable conformations, making them suitable materials in biomimetic structures and highly desirable matrices for the immobilization of bioreceptors, which play a key role in the development of biosensing devices [7,8].

Gold-based nanomaterials such as gold nanoparticles (AuNPs) have been widely used as surface modifiers for carbon and graphite electrodes, as their peculiar properties, coupled with their easy synthesis (e.g., applying a constant negative potential for a fixed time or by varying the potential for a different number of cycles at an optimal scan rate [9,10,11,12]) have attracted particular attention in their application for the development of sensors and biosensors [13,14]. Exploiting the high affinity between gold and thiol groups has in many cases enabled a further modification of these surfaces by immobilizing a plethora of thiolated biorecognition elements, such as antibodies, DNA strands, enzymes, etc. Moreover, the inclusion of AuNPs in conductive polymers can enhance electron transfer through a direct or mediated mechanism with improved conductivity and higher stability [15,16,17].

Taking into account all the above considerations, this paper presents the optimization of multiple nanocomposite platforms, based on PAAs and AuNPs, to be applied in the development of electrochemical sensors and biosensors. Graphite-based screen-printed electrochemical cells were modified by cyclic voltammetry (CV), and then the concentration of the precursors and the number of CV cycles for both modification steps (i.e., the electropolymerization of amino acids and the electrodeposition of AuNPs) were optimized. The resulting platforms were tested in the electrochemical oxidation of 1-naphthol, a redox molecule widely detected in electrochemical bioassays as an enzymatic product of phosphatases and thus chosen as a model analyte, by differential pulse voltammetry (DPV).

## 2. Materials and Methods

### 2.1. Chemicals

l-alanine (C_3_H_7_NO_2_), l-arginine hydrochloride (C_6_H_14_N_4_O_2_·HCl), l-aspartic acid (C_4_H_7_NO_4_), l-cysteine hydrochloride (C_3_H_7_NO_2_S·HCl), l-glutamic acid (C_5_H_9_NO_4_), l-glycine (C_2_H_5_NO_2_), l-histidine (C_6_H_9_N_3_O_2_), l-leucine (C_6_H_13_NO_2_), l-lysine hydrochloride (C_6_H_14_N_2_O_2_·HCl), l-proline (C_5_H_9_NO_2_), l-serine (C_3_H_7_NO_3_), l-tryptophan (C_11_H_12_N_2_O_2_), l-tyrosine (C_9_H_11_NO_3_), l-valine (C_5_H_11_NO_2_), tetrachloroauric acid (HAuCl_4_), sulphuric acid 97% (H_2_SO_4_), hydrochloric acid 37% (HCl), di-sodium hydrogen phosphate (Na_2_HPO_4_), sodium di-hydrogen phosphate di-hydrate (NaH_2_PO_4_·2H_2_O), sodium chloride (NaCl), potassium ferrocyanide (K_4_[Fe(CN)_6_]), potassium ferricyanide (K_3_[Fe(CN)_6_]), potassium chloride (KCl), 1-naphthol (C_10_H_8_O), diethanolamine (C_4_H_11_NO_2_), magnesium chloride hexahydrate (MgCl_2_·6H_2_O) were purchased from Merck (Milan, Italy). All chemicals were of analytical grade, and all solutions were prepared in deionized water (resistivity: 18 MΩ) from a Milli-Q system.

The buffer solutions used in this work were the following.

-Polymerization buffer: 0.1 M phosphate buffer pH 6.0, containing 0.1 M NaCl (PBS). The buffer was prepared by mixing a 0.1 M Na_2_HPO_4_ solution with a 0.1 M NaH_2_PO_4_ solution, each one containing 0.1 M NaCl, until the desired pH was reached.-Detection buffer: 0.1 M diethanolamine buffer pH 9.6, containing 0.1 M KCl and 1 mM MgCl_2_ (DEA). The buffer was prepared by adding 2 M HCl to a 0.1 M DEA solution, containing 0.1 M KCl and 2 mM MgCl_2_, until the desired pH was reached.

### 2.2. Apparatus

The electrochemical measurements were performed with a PalmSens2 and a Sensit Smart portable potentiostat/galvanostats (PalmSens BV, Houten, The Netherlands) controlled by PSTrace 5.11 software for data acquisition and processing. The nanocomposite electrochemical platforms were developed on screen-printed cells (SPCs), composed of a 3 mm-diameter graphite working electrode, a graphite counter electrode, and a silver pseudo-reference electrode on a plastic substrate (EcoBioServices and Researches, Sesto Fiorentino (FI), Italy). All the reported potentials refer to the pseudo-reference silver screen-printed electrode and all the measurements were carried out at room temperature.

Scanning electron microscopy (SEM) analysis was carried out using Gaia 3 microscope (Tescan a. s., Brno, Czech Republic). SEM images were acquired using an acceleration voltage of 5 kV for the bare graphite and of 10 kV for the nanostructured electrodes. Energy dispersive X-ray analysis (EDX) was performed to assess the elemental composition of the modified electrode surface.

### 2.3. Development of the Nanocomposite Electrochemical Platforms

The graphite-based screen-printed working electrodes were progressively modified by the electropolymerization of l-amino acids in phosphate buffer and then by the electrodeposition of gold nanoparticles in sulphuric acid. Each deposition step was optimized in terms of concentration of the precursor and number of CV cycles.

#### 2.3.1. Electropolymerization of l-Amino Acids

l-amino acids polymerization was performed by CV by dropping (50 μL) solutions of the monomers at different concentrations (1, 2, 5, 10 mM) in PBS onto the electrode surface and by scanning the potential between −1.5 V and +2.0 V at 100 mV/s. The modified electrodes were washed with Milli-Q water to remove excess monomers and free ions from the surface.

#### 2.3.2. Electrodeposition of Gold Nanoparticles

The polymerized surfaces were further modified by the electrodeposition of gold nanoparticles using CV. Solutions of HAuCl_4_ at different concentrations (0.1, 0.2, 0.5, 1 mM) in 0.5 M H_2_SO_4_ were dropped (50 μL) on the modified cells and the potential was scanned between −0.2 V and +1.2 V at 100 mV/s, as previously reported [18]. The AuNPs/p(l-AA)-modified SPCs were then washed with Milli-Q water to remove residual precursors and free ions from the surface and then stored under dry conditions.

### 2.4. Electrochemical Characterization of the Nanocomposite Platforms

To gain insights into the nanostructures on the electrode surface, and especially into their influence on the electrochemical performance of the screen-printed cells, an electrochemical characterization of the developed sensing platforms was performed by CV and electrochemical impedance spectroscopy (EIS) techniques in presence (50 μL) of a solution containing 5 mM [Fe(CN)_6_]^−4^ and 5 mM [Fe(CN)_6_]^−3^ redox probes in 0.1 M KCl. For CV measurements, the potential was scanned from −0.5 V to +0.8 V at different scan rates (25, 50, 75, 100, 125, and 150 mV/s). The current peak height (*i_p_*, in [A]) of both anodic and cathodic peaks was taken as the electrochemical signal and plotted vs. the square root of the scan rate (*v*^1/2^, in [V/s]). The obtained curves were fitted with the Randles–Sevcik equation [19]:*i_p_* = (2.69∙10^5^) *n*^3/2^
*A c* (*D v*)^1/2^(1)
where *n* is the number of electrons transferred in the redox event (=1, in the case of Fe(II)/Fe(III) couple), *A* (in [cm^2^]) is the electroactive surface area, *c* (in [mol/cm^3^]) is the probe bulk concentration, and *D* is the diffusion coefficient of the redox species (reported to be 6.67·10^−6^ cm^2^/s for [Fe(CN)_6_]^−4^ and 7.26·10^−6^ cm^2^/s for [Fe(CN)_6_]^−3^ [20]). For EIS measurements, the frequency was scanned in the range 100 kHz–10 mHz with an amplitude of 10 mV at a fixed DC potential of +0.13 V. After each CV measurement, the SPCs were discarded.

### 2.5. 1-Naphthol Electrochemical Detection

The electrochemical performance of the nanocomposite platforms was assessed by detecting 1-naphthol by means of DPV. Solutions at different concentrations of 1-naphthol (0–100 mg/L) in DEA buffer were dropped (50 μL) on different modified cells and the potential was scanned from −0.8 V to +0.8 V at 5 mV/s (pulse potential: 70 mV, pulse time: 50 ms). After each DPV measurement, the SPCs were discarded.

## 3. Results and Discussion

The presented nanocomposite platforms were built by combining a polymeric layer of poly(l-amino acids) and AuNPs with the dual purpose of improving the electrochemical performance of graphite-based SPCs and to provide a scaffold for the immobilization of bioreceptors by exploiting, for instance, the affinity between gold and thiol moieties.

The fourteen amino acids used in this work were selected because they are the most commonly used in sensors and biosensors development [7].

### 3.1. Optimization of Amino Acids Electropolymerization

The concentration of the monomer and the number of CV cycles were optimized for each l-amino acid by considering parameters such as the average of the anodic and cathodic current peak heights (*ī_p_*), their ratio (|*i_p_*_,__*a*_/*i_p_*_,__*c*_|), and the peak potential difference (Δ*E_p_*) related to the redox couple [Fe(CN)_6_]^−4/−3^. The results related to monomers’ concentration are shown in Figure 1 and Table 1.

The results related to CV cycles for the electropolymerization of amino acids are shown in Figure 2 and Table 2.

The appropriate monomer concentration of the monomers and the number of CV cycles for the electropolymerization were chosen by taking into account firstly the highest value of *ī_p_* and the closeness of |*i_p_*_,*a*_/*i_p_*_,*c*_| to 1, and secondly the lowest value of Δ*E_p_*, with the aim of selecting those polymerization conditions that provided (i) an increase in the electrochemical performance of the working electrode in oxidizing and reducing the redox probe and (ii) an improvement in the reversibility of the redox reaction. The chosen values are reported in Table 3.

To address a possible enhancement of the electrochemical properties of the SPCs brought by the modification of the graphite working electrodes with the sole PAAs films, EIS measurements were performed: the resulting spectra, presented in the form of complex plane diagrams (i.e., Nyquist plots), are shown in Figure 3.

From the reported plots, it can be seen that the diameter of the circular part of each graph, which is related to the charge transfer resistance (*R_ct_*), is lower for PAAs-modified surfaces than that of bare graphite, thus confirming that the electrochemical properties of SPCs are improved when a poly(amino acid) film is deposited onto the surface of the working electrodes. The only exception is the case of p(l-Trp), for which an increase in the *R_ct_* value can be observed with respect to bare graphite: this is probably due to a hampering effect of the side chain of l-Trp on the electron transfer.

### 3.2. Optimization of Gold Nanoparticles Electrodeposition

A similar optimization procedure was applied in choosing the concentration of HAuCl_4_ and the number of CV cycles related to the electrodeposition of AuNPs on each poly(l-amino acid)-modified SPC. The results related to HAuCl_4_ concentration are shown in Figure 4 and in Table 4.

The results related to CV cycles for the electrodeposition of gold nanoparticles are shown in Figure 5 and in Table 5.

The most appropriate parameters were chosen according to the same criteria explained in the previous section, and these are reported in Table 6.

### 3.3. Calculation of the Electroactive Surface Area

After having optimized the experimental parameters related to the electropolymerization of l-amino acids and the electrodeposition of AuNPs, an electrochemical characterization of the resulting nanocomposite platforms was performed by means of CV in the presence of the reversible redox couple [Fe(CN)_6_]^−4/−3^. From the angular coefficient of the linear regressions obtained by plotting *i_p_* vs. *v*^1/2^ for each platform, the electroactive surface area was calculated and compared with that of a bare graphite-based working electrode ((7.2 ± 0.4) mm^2^). The obtained values are shown in Table 7.

Considering the obtained results, it can be stated that a general increase in the electroactive surface area was observed for each nanocomposite platform compared to the bare graphite electrode, with an average of approximately 8.4 mm^2^. The only exceptions are those containing l-Arg, l-Glu and l-Lys (average ≈ 9.6 mm^2^), l-His and l-Ser (average ≈ 7.9 mm^2^), and l-Trp (4.2 mm^2^).

### 3.4. Morphological Characterization

The morphology of GSPEs was investigated before and after the electrochemical modification of the working electrodes with a poly(amino acid) and gold nanoparticles; moreover, to assess the effective modification of the graphite surface, EDX analysis of the nanostructured GSPEs was also carried out (Figure 6).

The results show (i) a smoothing effect given by the presence of the polymer and (ii) the presence of gold nanoparticles (i.e., the small white spots), randomly and homogeneously distributed onto the entire surface of the modified working electrode. These findings are confirmed by the EDX analysis, where a characteristic gold band around 2.1 keV can be seen when the poly(amino acid) and gold nanoparticles-modified screen-printed electrodes were investigated.

### 3.5. Electrochemical Performance of the Platforms Towards 1-Naphthol Oxidation

To evaluate the electrochemical performance of the resulting nanocomposite platforms, 1-naphthol was chosen as a model redox molecule and its detection was performed by DPV. This molecule is the enzymatic product of phosphatases, starting from 1-naphthyl phosphate as the enzymatic substrate. Phosphatases are widely used as labels with signal amplification purposes in biosensors and bioassays; thus, when the electrochemical transduction is exploited, as in this case, 1-naphthol is the molecule being electrochemically detected. Based on the above-mentioned considerations, the tested platforms were those with the highest electroactive surface areas, namely those containing l-Arg, l-Glu, and l-Lys. The resulting DPV scans and calibration curves are shown in Figure 7.

The oxidation current of 1-naphthol increases with increasing its concentration in all cases. However, regarding the shape of the voltammetric curves, the nanocomposite platform containing poly(l-Glu) produced more uniform peaks than those obtained on the platforms containing poly(l-Arg) and poly(l-Lys), which are characterized by a lower degree of symmetry and are represented by non-smooth lines. This lack of regularity is caused by the presence of multiple collateral shoulders and overlapping peaks, which was observed when 1-naphthol was detected on polymerized l-Arg and l-Lys. This could probably be due to the interaction of intermediate products, deriving from the electrooxidation of 1-naphthol, with the amino residues of l-Arg and l-Lys: in fact, arginine residues in proteins can be revealed in the presence of 1-naphthol [21], while 1-naphthol derivatives have the ability to selectively target lysine residues [22]. Moreover, the scans performed on the platform containing p(l-Glu) are also characterized by an enhanced reproducibility of the baseline among the different concentrations of 1-naphthol, which can probably be linked to the reproducibility of the modification processes themselves. Last but not least, the sensitivity (0.325 mA·L/g) and limit of detection (1.7 mg/L) retrieved on this platform are definitely better if compared to those obtained when p(l-Arg) (sensitivity: 0.207 mA·L/g, detection limit: 4.7 mg/L) or p(l-Lys) (sensitivity: 0.087 mA·L/g, detection limit: 17.1 mg/L) are present. To the best of our knowledge, 1-naphthol is usually detected on conventional solid electrodes, as glassy carbon [23,24,25] and carbon paste [26] ones, while only one detection example on screen-printed electrodes is reported [27]. Despite the extremely low detection limit attained (10 nM ≈ 1.4 μg/L), the presented graphene network would probably not be suitable for the purposes of our research, due to non-specific adsorption phenomena that may occur in the presence of biological entities (e.g., antibodies, enzymes, etc.). To further confirm the optimized performance of the platforms with the highest electroactive surface, DPV analysis was performed again to compare the effect of both monomers’ (l-Arg, l-Glu, l-Lys) concentrations and CV scan numbers on 1-naphthol detection, in the same way as presented in Section 3.1 and Section 3.2. The sensitivity values for 1-naphthol obtained under these conditions are reported in Table 8.

The obtained results for the detection of 1-naphthol show that the concentration and the number of CV cycles for the electropolymerization of the chosen l-AAs which yielded the highest sensitivity are in accordance with those presented in Table 3. Moreover, to definitively demonstrate the bio-affinity enhancement brought by the presence of the polymers on the nanocomposite platforms, 1-naphthol was also detected on AuNPs-modified graphite electrodes. The resulting DPV scans and calibration curve are shown in Figure 8.

The results show that the sensitivity towards 1-naphthol detection is lowered by approximately 60% with respect to the AuNPs/p(l-Glu) platform when only AuNPs are present on the electrode surface, thus demonstrating the enhancement of bio-affinity given by the presence of PAAs as electrode modifiers in combination with gold nanoparticles.

## 4. Conclusions

This paper presents the optimization of multiple electrochemical platforms, based on poly(l-amino acids) and gold nanoparticles, to be applied in the development of sensors and biosensors. Both the building steps of the platforms were optimized in terms of precursors’ concentrations and cyclic voltammetry (CV) scans. The optimized platforms were electrochemically characterized, and those with the highest electroactive surface areas (i.e., those containing l-Arg, l-Glu and l-Lys) were applied in the electrochemical detection of 1-naphthol, taken as a model analyte. Oxidation peaks were observed at the nanocomposite platforms, with the current peak height increasing with increasing 1-naphthol concentration. The developed platforms showed themselves to be suitable for applications in the development of sensors and biosensors, especially those involving architectures with 1-naphthol as the final detection product.

## Figures and Tables

**Figure 1 sensors-25-07230-f001:**
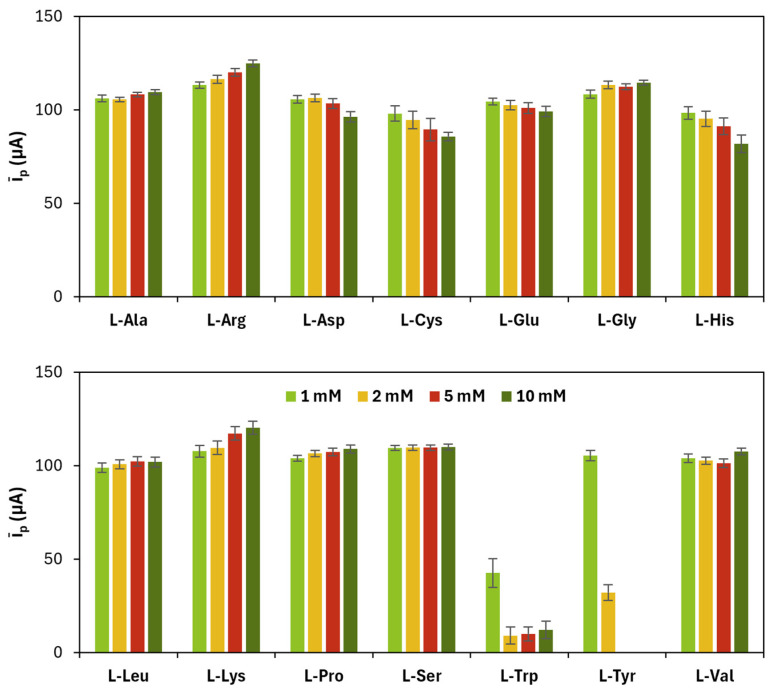
Average (*ī_p_*) of anodic (*i_p_*_,*a*_) and cathodic (*i_p_*_,*c*_) current peak heights obtained for each poly(l-amino acid)-modified SPC at different concentrations of the monomer (number of CV cycles: 10). The maximum solubility for l-Tyr is 2 mM.

**Figure 2 sensors-25-07230-f002:**
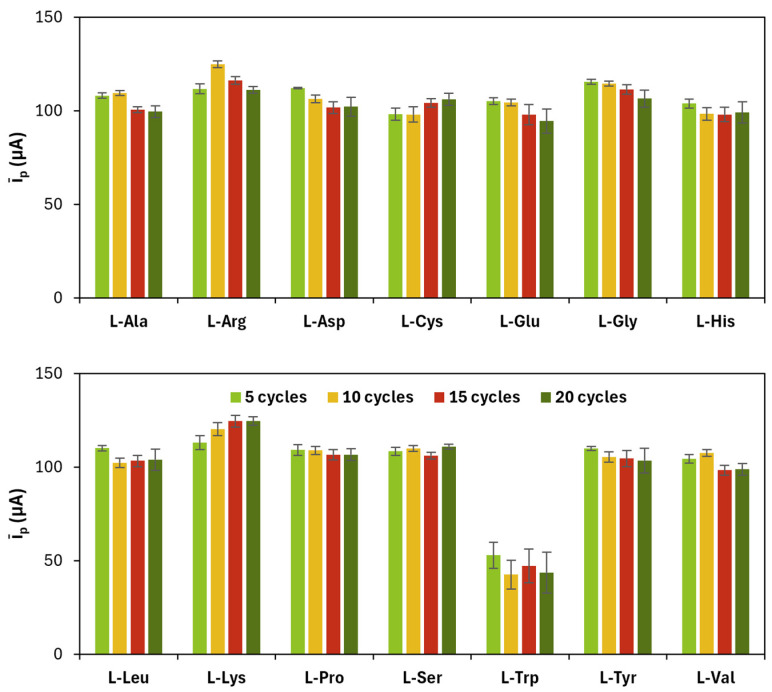
Average (*ī_p_*) of anodic (*i_p_*_,*a*_) and cathodic (*i_p_*_,*c*_) current peak heights obtained for each poly(l-amino acid)-modified SPC at a different number of CV cycles.

**Figure 3 sensors-25-07230-f003:**
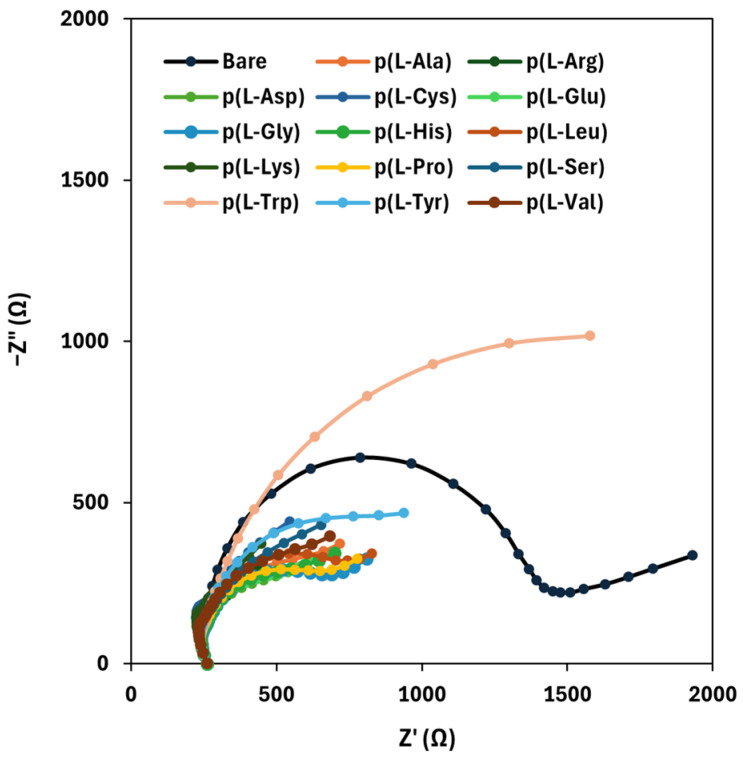
Nyquist plots of bare and PAAs-modified graphite electrodes.

**Figure 4 sensors-25-07230-f004:**
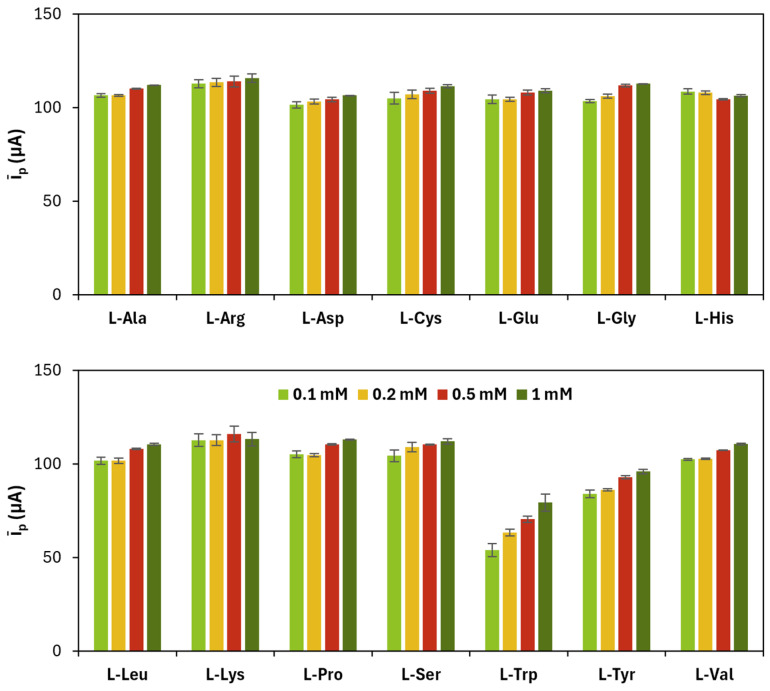
Average (*ī_p_*) of anodic (*i_p_*_,*a*_) and cathodic (*i_p_*_,*c*_) current peak heights obtained for each gold nanoparticles and poly(l-amino acid)-modified SPC at different concentrations of the precursor (number of CV cycles: 15).

**Figure 5 sensors-25-07230-f005:**
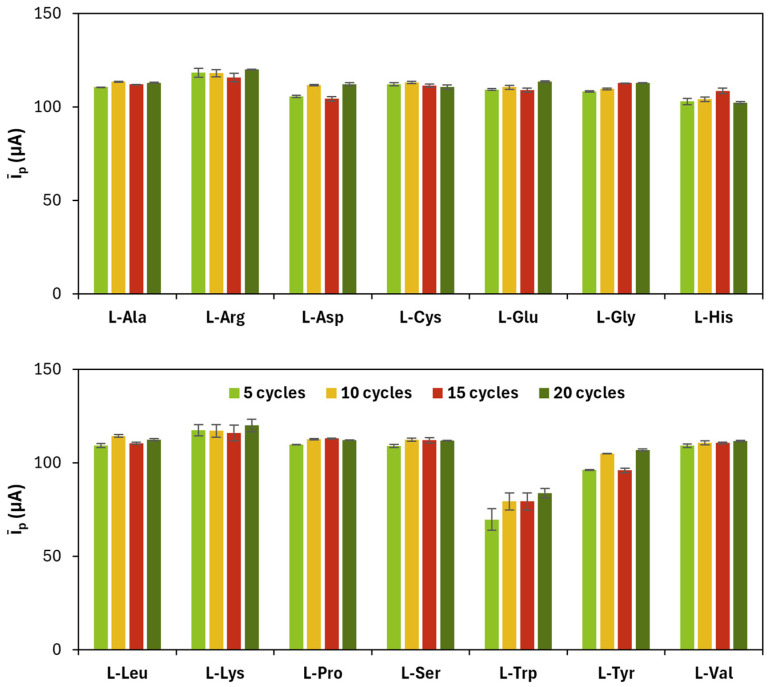
Average (*ī_p_*) of anodic (*i_p_*_,*a*_) and cathodic (*i_p_*_,*c*_) current peak heights obtained for each gold nanoparticles and poly(l-amino acid)-modified SPC at different number of CV cycles.

**Figure 6 sensors-25-07230-f006:**
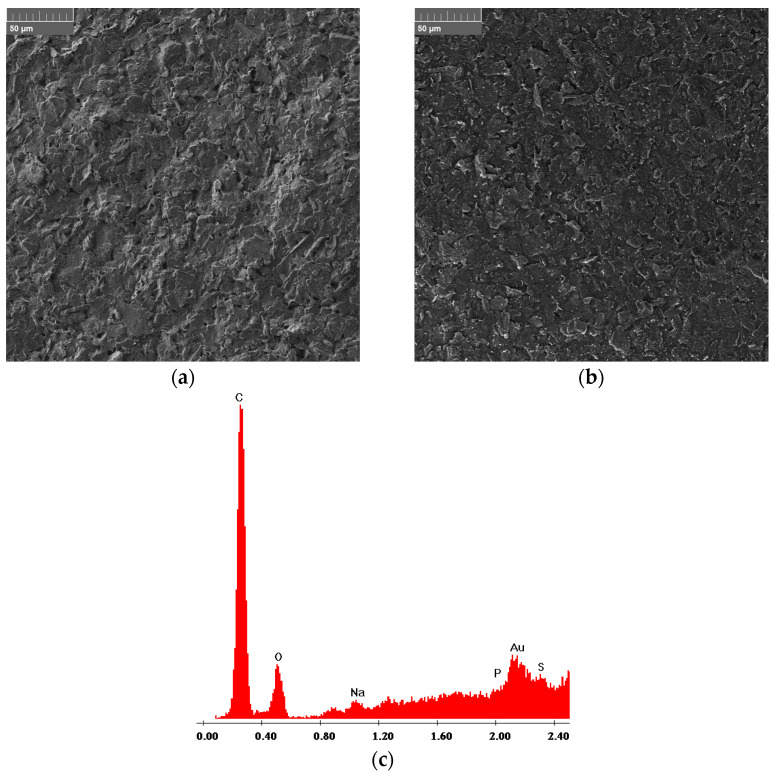
SEM morphologies of (**a**) bare and (**b**) AuNPs/pAA-modified graphite electrodes and (**c**) EDX spectrum of nanocomposite-modified GSPEs. A representative sample containing l-Glu as the amino acid was analyzed.

**Figure 7 sensors-25-07230-f007:**
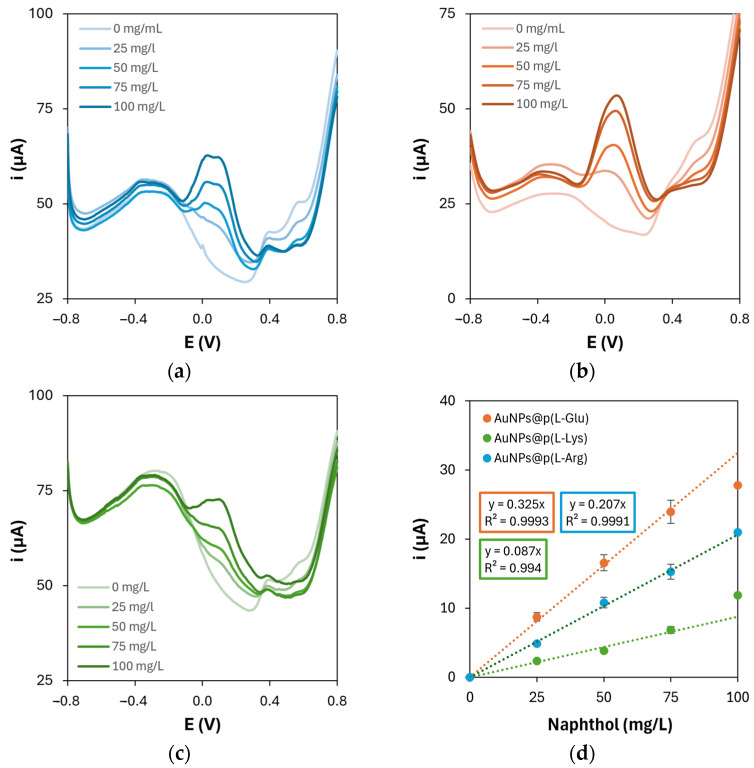
Differential pulse voltammograms of 1-naphthol oxidation on gold nanoparticles and (**a**) poly(l-Arg)-, (**b**) poly(l-Glu)-, or (**c**) poly(l-Lys)-modified graphite electrodes; (**d**) corresponding calibration curves.

**Figure 8 sensors-25-07230-f008:**
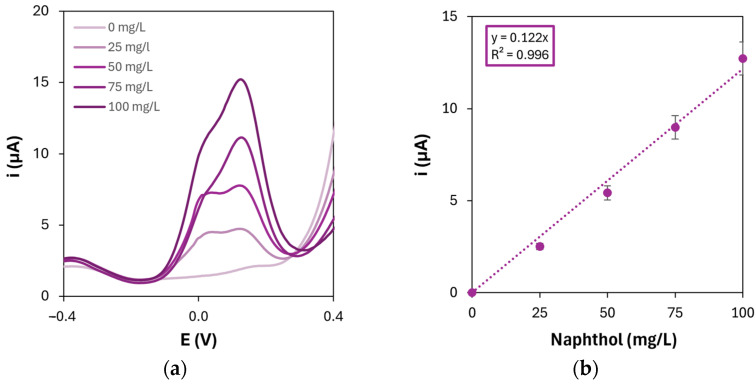
(**a**) Differential pulse voltammograms of 1-naphthol oxidation on gold nanoparticles-modified graphite electrodes and (**b**) corresponding calibration curve ([HAuCl_4_] = 1 mM; number of CV cycles for the electrodeposition: 20).

**Table 1 sensors-25-07230-t001:** List of ratios between anodic and cathodic current peak heights (**|*i_p_*_,*a*_*/i_p_*_,*c*_|**) and peak potential differences (Δ*E_p_*) obtained for each poly(l-amino acid)-modified SPC at different concentrations of the monomer (number of CV cycles: 10).

	|i_p,a_/i_p,c_|				ΔE_p_ (mV)			
[l-AA] (mM)	1	2	5	10	1	2	5	10
l-Ala	0.98	0.98	0.98	0.98	260	255	260	250
l-Arg	0.98	0.98	0.98	0.98	235	225	235	225
l-Asp	0.97	0.97	0.97	0.96	225	225	230	240
l-Cys	0.94	0.93	0.91	0.96	195	200	200	220
l-Glu	0.98	0.97	0.96	0.96	230	230	235	255
l-Gly	0.97	0.97	0.98	0.98	220	225	220	215
l-His	0.95	0.94	0.93	0.92	215	215	210	195
l-Leu	0.96	0.97	0.97	0.96	250	240	245	240
l-Lys	0.96	0.95	0.96	0.96	205	205	200	185
l-Pro	0.98	0.98	0.97	0.97	260	250	240	230
l-Ser	0.98	0.98	0.98	0.98	235	235	230	235
l-Trp	0.77	0.47	0.58	0.56	205	85	70	70
l-Tyr *	0.96	1.21	-	-	205	490	-	-
l-Val	0.97	0.97	0.97	0.98	220	230	220	215

* The maximum solubility of l-Tyr in H_2_O is 2 mM.

**Table 2 sensors-25-07230-t002:** List of ratios between anodic and cathodic current peak heights (**|*i_p_*_,*a*_*/i_p_*_,*c*_|**) and peak potential differences (Δ*E_p_*) obtained for each poly(l-amino acid)-modified SPC at different number of CV cycles.

	|i_p,a_/i_p,c_|				ΔE_p_ (mV)			
CV Cycles	5	10	15	20	5	10	15	20
l-Ala (10 mM)	0.98	0.98	0.98	0.96	235	250	265	280
l-Arg (10 mM)	0.97	0.98	0.98	0.98	215	225	245	260
l-Asp (2 mM)	0.99	0.97	0.96	0.93	240	225	250	270
l-Cys (1 mM)	0.96	0.94	0.97	0.96	210	195	210	220
l-Glu (1 mM)	0.98	0.98	0.92	0.91	220	230	250	275
l-Gly (10 mM)	0.98	0.98	0.97	0.94	205	215	230	265
l-His (1 mM)	0.97	0.95	0.95	0.92	215	215	225	230
l-Leu (5 mM)	0.98	0.97	0.96	0.93	225	245	250	275
l-Lys (10 mM)	0.95	0.96	0.97	0.97	195	185	200	215
l-Pro (10 mM)	0.96	0.97	0.96	0.96	225	230	240	270
l-Ser (10 mM)	0.97	0.98	0.98	0.98	235	235	250	255
l-Trp (1 mM)	0.83	0.77	0.76	0.70	240	205	200	205
l-Tyr (1 mM)	0.98	0.96	0.94	0.91	205	205	230	265
l-Val (10 mM)	0.97	0.98	0.96	0.95	215	215	220	235

**Table 3 sensors-25-07230-t003:** Optimized parameters for the polymerization of l-amino acids on SPCs.

	[l-AA] (mM)	CV Cycles
l-Ala	10	10
l-Arg	10	10
l-Asp	2	5
l-Cys	1	20
l-Glu	1	5
l-Gly	10	5
l-His	1	5
l-Leu	5	5
l-Lys	10	15
l-Pro	10	5
l-Ser	10	20
l-Trp	1	5
l-Tyr	1	5
l-Val	10	10

**Table 4 sensors-25-07230-t004:** List of ratios between anodic and cathodic current peak heights (**|*i_p_*_,*a*_/*i_p_*_,*c*_|**) and peak potential differences (Δ*E_p_*) obtained for each gold nanoparticles and poly(l-amino acid)-modified SPC at different concentrations of the precursor (number of CV cycles: 15).

	|i_p,a_/i_p,c_|				ΔE_p_ (mV)			
[HAuCl_4_] (mM)	0.1	0.2	0.5	1	0.1	0.2	0.5	1
l-Ala (10 mM)	0.99	0.99	1.00	1.00	235	235	235	235
l-Arg (10 mM)	0.97	0.97	0.97	0.97	220	210	210	210
l-Asp (2 mM)	0.98	0.98	0.98	1.00	210	210	205	215
l-Cys (1 mM)	0.96	0.97	0.98	0.99	225	235	230	225
l-Glu (1 mM)	0.97	0.98	0.98	0.99	210	215	215	220
l-Gly (10 mM)	0.99	0.99	0.99	1.00	215	220	215	220
l-His (1 mM)	0.98	0.99	1.00	1.01	210	210	220	225
l-Leu (5 mM)	0.97	0.98	0.99	0.99	245	245	230	230
l-Lys (10 mM)	0.96	0.96	0.95	0.96	220	225	220	245
l-Pro (10 mM)	0.97	0.99	1.00	1.00	240	240	225	230
l-Ser (10 mM)	0.96	0.97	1.00	0.98	235	235	230	245
l-Trp (1 mM)	0.91	0.96	1.03	1.09	270	260	290	270
l-Tyr (1 mM)	0.97	0.99	0.99	0.98	235	235	235	235
l-Val (10 mM)	0.99	1.00	1.00	1.01	225	230	235	235

**Table 5 sensors-25-07230-t005:** List of ratios between anodic and cathodic current peak heights (**|*i_p_*_,*a*_/*i_p_*_,*c*_|**) and peak potential differences (Δ*E_p_*) obtained for each gold nanoparticles and poly(l-amino acid)-modified SPC at different number of CV cycles.

		|i_p,a_/i_p,c_|				ΔE_p_ (mV)			
CV Cycles		5	10	15	20	5	10	15	20
l-Ala (10 mM)	HAuCl_4_ (1 mM)	1.00	1.00	1.00	1.00	235	235	235	230
l-Arg (10 mM)	HAuCl_4_ (1 mM)	0.97	0.98	0.97	1.00	205	210	210	210
l-Asp (2 mM)	HAuCl_4_ (0.5 mM)	0.99	0.99	0.98	1.01	215	210	205	210
l-Cys (1 mM)	HAuCl_4_ (1 mM)	0.99	0.99	0.99	1.01	230	230	225	245
l-Glu (1 mM)	HAuCl_4_ (1 mM)	0.99	0.99	0.99	0.99	215	215	220	225
l-Gly (10 mM)	HAuCl_4_ (1 mM)	1.00	1.01	1.00	1.00	215	220	220	210
l-His (1 mM)	HAuCl_4_ (0.1 mM)	0.98	0.98	0.98	0.99	220	225	210	230
l-Leu (5 mM)	HAuCl_4_ (1 mM)	0.99	0.99	0.99	1.01	230	220	230	235
l-Lys (10 mM)	HAuCl_4_ (0.5 mM)	0.96	0.96	0.95	0.96	215	220	220	235
l-Pro (10 mM)	HAuCl_4_ (1 mM)	1.00	1.00	1.00	1.00	230	225	230	225
l-Ser (10 mM)	HAuCl_4_ (1 mM)	0.99	1.01	0.98	1.00	240	240	245	255
l-Trp (1 mM)	HAuCl_4_ (1 mM)	1.12	1.08	1.09	1.04	300	270	270	270
l-Tyr (1 mM)	HAuCl_4_ (1 mM)	1.00	1.00	0.98	0.99	240	220	235	220
l-Val (10 mM)	HAuCl_4_ (1 mM)	1.01	1.01	1.01	1.01	230	230	235	225

**Table 6 sensors-25-07230-t006:** Optimized parameters for the deposition of gold nanoparticles on poly(l-amino acids)-modified SPCs.

	[HAuCl_4_] (mM)	CV Cycles
l-Ala	1	10
l-Arg	1	20
l-Asp	0.5	10
l-Cys	1	10
l-Glu	1	20
l-Gly	1	15
l-His	0.1	15
l-Leu	1	10
l-Lys	0.5	20
l-Pro	1	15
l-Ser	1	10
l-Trp	1	20
l-Tyr	1	20
l-Val	1	20

**Table 7 sensors-25-07230-t007:** Electroactive surface areas of gold nanoparticles and poly(l-amino acids)-modified SPCs.

	A_anodic_ (mm^2^)	A_cathodic_ (mm^2^)	A_average_ (mm^2^)	SD (mm^2^)	%RSD (%)
l-Ala	9.1	8.5	8.8	0.4	4.7
l-Arg	9.8	10.1	10.0	0.2	2.1
l-Asp	8.6	8.5	8.5	0.1	1.0
l-Cys	8.9	8.6	8.8	0.2	2.2
l-Glu	9.8	9.0	9.4	0.5	5.6
l-Gly	8.7	8.4	8.6	0.2	2.5
l-His	7.8	7.8	7.8	<0.1	0.2
l-Leu	8.6	8.4	8.5	0.2	2.1
l-Lys	9.4	9.7	9.5	0.3	2.6
l-Pro	8.7	8.7	8.7	<0.1	0.1
l-Ser	8.1	8.1	8.1	<0.1	0.1
l-Trp	4.8	3.6	4.2	0.9	20.7
l-Tyr	9.1	8.5	8.8	0.5	5.1
l-Val	8.8	8.3	8.5	0.3	4.0

**Table 8 sensors-25-07230-t008:** List of sensitivity values obtained for gold nanoparticles and p(l-Arg)-, p(l-Glu)-, and p(l-Lys)-modified SPCs at different concentrations of the monomers and number of CV cycles for the electropolymerization ([HAuCl_4_] = 1 mM for p(l-Arg) and p(l-Glu), [HAuCl_4_] = 0.5 mM for p(l-Lys); number of CV cycles for the electrodeposition: 20).

	Sensitivity (mA·L/g)			
**[l-AA] (mM)**	**1**	**2**	**5**	**10**
l-Arg	0.180	0.193	0.199	0.207
l-Glu	0.323	0.317	0.312	0.306
l-Lys	0.076	0.077	0.082	0.084
**CV cycles**	**5**	**10**	**15**	**20**
l-Arg (10 mM)	0.185	0.207	0.192	0.184
l-Glu (1 mM)	0.325	0.323	0.303	0.290
l-Lys (10 mM)	0.079	0.084	0.087	0.087

## Data Availability

The original contributions presented in this study are included in the article. Further inquiries can be directed to the corresponding author.

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
