# Peer review of "Optimization of Poly(l-Amino Acids)-Based Platforms for Sensing and Biosensing: A Cyclic Voltammetry Study"

_sensors, 2025, doi:10.3390/s25237230_

Round 1

Reviewer 1 Report (Previous Reviewer 1)

Comments and Suggestions for Authors

The revisions fully address the concerns raised during peer review, and the manuscript meets the standards for publication in Sensors.

Author Response

Reviewer 2 Report (Previous Reviewer 2)

Comments and Suggestions for Authors

I thank the authors for addressing my concerns in the previous revisions. In this revision, I have the following suggestions and recommendations.

line 235: i think the authors meant figure 6?

line 239: the authors mention "the presence of gold nanoparticles, randomly and homogeneously distributed", which i assume to be referring to the white specks visible on figure 6b. If so, they should describe it in the text, as currently it is up to the reader to interpret where the evidence is for the presence of gold nanoparticles. They could consider performing EDX analysis on the SEM instrument to experimentally demonstrate the presence of gold nanoparticles.

Comments on the Quality of English Language

line 174: "oxidating" should be "oxidizing"/"oxidising"

"table 6" in line 213 should be on line 212.

line 266: do the authors actually mean "properly" or did they mean "probably"?

line 280: i suggest "definitively" rather than "definitely"

reference 24: title should be changed to be sentence case rather than all uppercases to align with all the other references

Author Response

Reviewer 3 Report (Previous Reviewer 3)

Comments and Suggestions for Authors

The paper "Optimization of Poly(L-Amino Acids)-Based Platforms for Sensing and Biosensing: A Cyclic Voltammetry Study " is improved version of previously rejected manuscript. The paper presents steps for developing electrochemical platform based on the polymerized amino acids and deposited gold nanoparticles.

All previous remarks are addressed appropriately in new version of manuscript.

Additionally, author added new results, namely SEM and EIS investigation, and in that manner improved the quality of the manuscript.

Furthermore, the author improved visual presentation of the results.

The manuscript can be published.

Author Response

Reviewer 4 Report (Previous Reviewer 4)

Comments and Suggestions for Authors

The figures and data presentation have been significantly improved compared to the previous version of the manuscript. However, there is a huge scope for further improvement in this report. The comments or suggestions are:

  1. The authors completely omitted the detection mechanism in the manuscript. I strongly believe that the inclusion of a detection mechanism or peak current enhancement will help the readers to understand the presented work.
  2. 1-Naphthol is used as an analyte of interest, but the report doesn't include sufficient discussion on LoD and references to prove the enhancement and importance of this analyte.
  3. Authors must include other characterizations, such as Raman and FTIR, and their discussions for the modified electrode to draw the detection mechanism.

Round 2

Reviewer 2 Report (Previous Reviewer 2)

Comments and Suggestions for Authors

I thank the authors for making the changes and would recommend this revision for publication.

This manuscript is a resubmission of an earlier submission. The following is a list of the peer review reports and author responses from that submission.

Round 1

Reviewer 1 Report

Comments and Suggestions for Authors

In the manuscript entitled “Optimization of Poly(L-Amino Acids)-Based Platforms for Sensing and Biosensing: A Cyclic Voltammetry Study”, the authors report on the optimization of poly(L-amino acid) (PLA) coating on AuNP-graphite electrodes for electrochemical sensing applications.

The principal contributions of this work are: (1) comparison of various L-amino acid monomers in terms of redox reaction performance; (2) optimization of coating parameters including monomer concentration and CV cycle; and (3) demonstration of enhanced bio-affinity following PLA coating, evaluated via differential pulse voltammetry (DPV). This study introduces a potentially valuable approach for improving the bio-affinity property of biosensors through optimized PLA coating.

The work is well-motivated and thoughtfully designed. However, several aspects of the experimental methodology and data presentation require clarification and enhancement before the work can be considered for publication in Sensors. I recommend major revision to address the following points:

  1. The authors stated the dropping solutions (line 96, page 3) and the HAuCl4 (line 102, page 3). But the experimental conditions are not sufficiently described. The concentrations, solvent types, and preparation methods of all solutions should be clearly specified to ensure reproducibility.
  2. Tables 1, 2, 4, and 5 provide valuable optimization data for electrochemical performance; however, the readability could be significantly improved. I strongly recommend supplementing these tables with appropriately formatted plots or graphical representations to enhance data interpretation for readers.
  3. In tables 2 and 5, the concentration of each poly(L-amino acid) monomer should be explicitly stated.
  4. The manuscript mentions an increase in conductivity (line 150, page 5); however, conductivity cannot be conclusively inferred from CV curves alone. To substantiate this claim, a four-probe conductivity measurement is recommended, with comparative data presented for each experimental condition.
  5. In Figure 1, the concentration corresponding to each graph should be clearly indicated. Additionally, to convincingly demonstrate bio-affinity enhancement, the PLA-coated electrodes should be directly compared to the control (AuNP-graphite electrodes without PLA coating).
  6. The statement that “multiple collateral shoulders and/or overlapping peaks could be observed” (lines 192–193, page 8) requires further elaboration. The authors should explain the physical or electrochemical significance of these features, including possible causes and implications for sensor performance.
  7. The phrase “more uniform peaks” (line 191, page 8) is too general to convey meaningful interpretation. Please provide a more detailed explanation of what constitutes “uniformity” in this context—e.g., peak symmetry, reproducibility, or stability across cycles.
  8. The authors optimized the electrochemical performance of PLA-coated electrodes depending on monomer concentration and CV scan number. However, DPV analyses were performed only by varying 1-naphthol concentration (lines 202-204, page 8). To definitely validate the claim of enhanced bio-affinity, Figure 1 should include DPV results comparing the effects of both monomer concentration and CV scan number under consistent conditions.

Reviewer 2 Report

Comments and Suggestions for Authors

The authors have described a process of optimizing the fabrication of electrochemical devices comprising poly(amino acids) as well as gold nanoparticles. Through extensively varying the concentration and number of scan cycles, they identified the best combination and subsequently characterized its electrochemical performance using 1-napthol. The study is highly detailed and I have the following recommendations for the authors to improve their paper.

Line 35: the authors mention "these films can significantly improve the conductivity", but poly(amino acids) should generally be insulators? They cited reference 4 which makes a similar claim, but ref 4's corresponding reference 30,31 and 32 mentioned the conductivity change in relation to either nanoparticles or carbon nanotubes, none of which are present in PAA. In fact, https://doi.org/10.1016/j.jpha.2019.04.001  mentions "This larger resistance is attributed to the low electrical conductivity of the poly(Met) film, which hampers the electronic transfer on the electrode surface". The authors should reassess this statement, or perform a simple resistance measurement to support/clarify their claim.

Line 111: The authors mentioned using an equimolar concentration of  [Fe(CN)6]−4/−3 redox probe. Does this mean they used 2.5 mM of each since the total was 5 mM? I suggest rephrasing the statement to be more definitive.

In equation 1, the authors should make sure that their formatting should match what they used in the main text in terms of font and italicisation.

What volume of solutions did the authors add to the SPE during each step? They have only mentioned concentrations. Also, how did they ensure that the volume they added stayed in the desired position? Did they use some kind of container/well?

Table 1 and 2 could be reversed, since line 136 mentions number of cycles before monomer concentration.

line 155: extra charge sign in "HAuCl4"

Tables 4 and 5: why are some row entries more precise than the others with 2 decimal places? are different instruments used? Please also consider shading either the ip or ΔE column, or adding a vertical border in order to better differentiate the two; currently the column for ip is very wide and challenging to read.

line 179: the authors were inconsistent with reporting values from table 7, where came from different columns. Please justify this choice in the text or report the values from the same column.

Figure 1 needs a legend to see how the colours correspond to concentrations of 1-napthol used. The individual plots are better labelled with a, b, and c on the top left corner rather than below the horizontal axis. A more quantitative analysis is also preferred, such as performing a linear regression on the data and seeing which composite gives the greatest performance.

Line 194: It would be beneficial for the authors to propose a reason for the additional shoulder peaks for L-Arg ad L-Lys.

Some form of surface characterization might be helpful in quantifying and supporting the author's data on electroactive surface area. I would suggest attempting either SEM or AFM, as that would also show how the electrode surface changes with each fabrication step. They may perform this on a representative sample.

Since the authors used the Randles-Sevcik equation to calculate the electroactive surface area, they should mention what values they chose for the constant variables n, D and v.

Comments on the Quality of English Language

The authors' command of the english language is excellent and I suggest the following minor edits to improve clarity.

line 59: "was performed" might be better as "were performed" since two parameters were optimized.

line 85: "composed by a" should be "composed of a"

line 108: "nanostructuration" seems like an uncommon word and is better replaced with "nanostructure"

line 133: "as those most applied in" could be clearer as "as those are commonly used in"

line 137: "considering as the discriminating parameters" would be clearer as "considering parameters such as"

Reviewer 3 Report

Comments and Suggestions for Authors

The paper "Optimization of Poly(L-Amino Acids)-Based Platforms for Sensing and Biosensing: A Cyclic Voltammetry Study " presents steps for developing an electrochemical platform based on the polymerized amino acids and deposited gold nanoparticles. The approach taken in this study was strictly electrochemical, i.e. only electrochemical procedures and results, without characterization techniques for modified SPE. The paper had a good start, but it lacks scientific rigour, necessary for the successful presentation of the results obtained in this analytical approach.

The detailed remarks are given in the order of their appearance in the text:

1 – Abstract: word peculiar

Please check if this is the word you really want to use (line 17, line 43). According to the dictionary the meaning of word peculiar (adjective) is STRANGE, unusual and strange, sometimes in an unpleasant way.

2 – Section 2.4 and results related to it:

The authors stated that they used the Randles-Sevcik equation to fit the obtained curves.

  1. a) Please state what the reason was for this fitting - what result was obtained?
  2. b) Why did the authors use the Randles-Sevcik equation for the reversible system, since the data presented in Table 1 -5 shows peak-to-peak separation values above 200 mV, which is considered a margin for the irreversible systems?

Having this in mind, the results presented in Table 7 are questionable.

  1. c) The caption for Tables 1-5 only states that average current peak heights are presented. Average of what – several measurements or average of anodic and cathodic current? Again, the ratio between anodic and cathodic current is also important for establishing the reversibility of the redox process.
  2. d) What values of diffusion coefficient were used? Were those tabular values, or did the authors determine the values of diffusion coefficients?
  3. e) Since the authors calculated ECSA from anodic and cathodic current it is not correct to use the diffusion coefficient for the oxidised analyte (line 120). For anodic currents diffusion coefficient of oxidised analyte (i.e. the analyte that is going to be oxidised, in this case [Fe(CN)6]4-) should be used, and for cathodic currents diffusion coefficient of the reduced analyte should be used. Again, this raises a question about the results presented in Table 7.

3- Line 183: The authors stated that 1-naphthol was chosen as a model redox molecule

Please specify the rationale for selecting 1-naphthol as a model redox molecule and indicate the process or analyte type it represents.

4 – Section 3.4:

  1. a) What is the meaning of the different colours in Figure 1?
  2. b) The DPV technique is usually used to obtain a more sensitive determination of the analyte, preceded by cyclic voltammetry to determine at least the potential range. Were the parameters used for DPV optimized? If not, then how were DPV parameters selected?
  1. c) What could be the explanation for multiple shoulders obtained for poly(L-Arg) and poly(L-Lys)?

5 – Conclusion, line 202: Redox peaks were observed

Redox implies reduction-oxidation. Only oxidation peaks are visible in DPV.

Since the presented comments raise a serious question about results presented in this paper, the recommendation for the paper is to be rejected, with possible resubmitting after resolving the presented issues. 

Reviewer 4 Report

Comments and Suggestions for Authors

The manuscript is poorly written and lacks a lot of information. Authors should work on the figures to represent the different aspects of the presented work.
